# Inline Inspection of Packaged Food Using Microwave/Terahertz Sensing—An Overview with Focus on Confectionery Products

Mohieddine Jelali *⬤ and Konstantinos Papadopoulos ⬤

Cologne Laboratory of Artificial Intelligence and Smart Automation (CAISA), Institute of Product Development and Engineering Design (IPK), Technische Hochschule Köln—University of Applied Sciences, 50679 Cologne, Germany; konstantinos.papadopoulos@th-koeln.de
* Correspondence: mohieddine.jelali@th-koeln.de

**Abstract:** Electromagnetic systems, in particular microwave/terahertz sensing technologies, are the newest among nondestructive sensing technologies. Currently, increased attention is pointed towards their use in various applications. Among these, food inspection stands out as a primary area due to its potential risk to human safety. As a result, substantial efforts are currently focused on utilizing microwave/terahertz imaging as a tool to enhance the efficacy of food quality assessments. This paper deals with the exploitation of microwave/terahertz imaging technology for food quality control and assessment. In particular, the work aims at reviewing the latest developments regarding the detection of internal quality parameters, such as foreign bodies, i.e., plastic, glass, and wood substances/fragments, as well as checking the completeness of the packaged food under consideration. Emphasis is placed on the (inline) inspection of wrapped/packaged food, such as chocolates, cookies, pastries, cakes, and similar confectionery products, moving along production conveyor belts. Moreover, the paper gives a recent overview of system prototypes and industrial products and highlights emerging research topics and future application directions in this area.

**Keywords:** packaged food inspection; foreign body detection; completeness check; electromagnetic sensing; radar/terahertz imaging





## 1. Introduction

Ensuring the quality and safety of food is essential for public health, social stability and development. Food safety covers a broad spectrum, including concerns related to physical, chemical and biological contamination, as well as other associated hazards or toxins. Physical contamination of food is a major concern, particularly faecal contamination and defects closely associated with harmful pathogens or irregularities. It also includes the presence of foreign materials such as stones from the environment, metals, glass from equipment, fingernails from processors and bone fragments. These foreign objects can enter food at various stages, including harvesting, processing and final packaging [1].

According to Nestlé, one of the world's largest food manufacturers, the prevention and detection of foreign objects is an important element of food safety and compliance. A foreign body is any material such as metal, glass, plastic, stones, etc., animal origin (insects, bones, hair, etc.), plant origin (wood stalks, etc.) or product origin (burnt particles, scorched particles, etc.) that consumers do not want or expect to find in the products they buy [2]. Usually, food manufacturers are liable for contamination, so foreign objects in their food products can lead to a bad reputation and loss of sales, and contaminants can pose health risks to consumers (e.g., injury and choking). For example, unwrapping a bar of chocolate may reveal an unpleasant surprise such as an insect, pieces of metal or glass, or clumps of dirt. This can not only spoil the consumer's appetite, but also cause a broken tooth or poisoning [3].

Accurate detection of food defects is critical to the drive for improved quality and safety. Manual inspection of food defects alongside conveyor systems has been the norm,

but its efficiency declines over time. As a result, many automated non-destructive testing (NDT) techniques have been used for accurate inline food quality control. These include metal detection, X-ray, acoustic emission, ultrasound, thermal imaging, hyperspectral/multispectral imaging, fluorescence spectroscopy, radar/terahertz imaging and near-infrared (NIR) spectroscopy. Reviews on this topic are reported in [4–7].

The advantages and disadvantages of the most common NDT methods used to check the quality of food products, i.e., to detect internal defects and foreign bodies, can be summarised as follows:

- **Metal detectors** are a common feature in food processing facilities, often acting as the final barrier to foreign material before packaging. These systems are primarily designed to automatically detect and isolate metal contaminants. However, metal is only one facet of the potential contamination risks. There are many other hazards such as rubber, glass, hard plastics, seeds, insect bodies and non-magnetic contaminants that require complementary or alternative methods to ensure food safety and quality [6].

- **X-ray systems**, now increasingly used in the food industry for quality control inspections, offer exceptional image resolution but face challenges in identifying low-density objects such as plastic, glass, wood or insects. In addition, the use of ionising radiation is associated with inherent risks to both operators and the food product itself, potentially altering its properties [8].

- **Infrared (IR) technologies** boast speed and safety as key advantages, but are limited by limited penetration and significant absorption in water. Conversely, fluorescence imaging is only effective when examining objects containing fluorescent compounds [8].

- **Near-Infrared (NIR) Spectroscopy** offers many advantages, including its non-ionising nature, its ability to penetrate air gaps within food materials and its ability to detect small elements within the internal structure of food. However, NIR spectroscopy calculates an average spectrum of a sample, providing a single spectrum, but the resulting data can be insufficient and complex for analysis. In addition, NIR has limitations due to its reliance on reference methods for calibration [6].

- **Thermal imaging systems**, comprising a camera, optical components (such as a focusing lens, collimating lenses and filters), a detector array, signal processing and image processing systems, offer real-time operation without emitting harmful radiation. However, their use in the food industry is limited by temperature interference from other surfaces [6,9].

In this paper, we consider electromagnetic (EM) systems as an efficient and rapid solution for the detection of contaminants or foreign objects due to their non-invasive nature and ability to penetrate various types of food. One of the key advantages is their ability to identify foreign objects regardless of the packaging material, allowing for a comprehensive inspection of packaged goods without compromising product integrity. EM-based inspection technology is also preferred because of its insensitivity to adverse conditions in production environments, such as high temperatures, dust, humidity and water vapour/mist. Due to the use of non-ionising radiation, today's EM systems do not require any preventive measures for occupational safety (in contrast to laser technology or radiometry). Another strength is the privacy of EM imaging solutions (unlike camera-based systems).

The contributions of this paper are as follows:

- It focuses on methods for the *detection of internal quality parameters such as foreign bodies*, in particular, plastic, glass and wood substances/fragments, as well as *checking for completeness* of the packaged food under consideration. We also limit ourselves to the (inline) inspection of (non-metallic) packaged food products such as chocolates, biscuits, pastries, cakes, and similar confectionery products moving along production conveyor belts. In this context, we are looking at EM sensing technologies as a novel, non-destructive, accurate and safe option. To the best of our knowledge, this is the first systematic review of its kind.

- It provides an up-to-date overview of system prototypes and industrial products to guide researchers and practitioners in advancing or selecting the right technology for their use cases.
- It evaluates the related work based on critical aspects of integration in industrial applications
- It highlights several emerging research topics and future application directions in the field.

The rest of the paper is organised as follows: For the sake of completeness, Section 2 briefly describes the main principles of NDT based on microwave/terahertz sensing/imaging. Then, in Section 3, related work with a focus on confectionery products is reviewed, analysed and classified. Section 4 focuses on presenting an overview of NDT prototypes and industrial products, Section 5 on discussing some aspects of industrial suitability, practicality, reliability, and accuracy of the products and applications. Finally, the main open challenges and future perspectives are discussed in Section 6.

## 2. Classification and Principles of Non-Destructive Testing Based on Microwave/Terahertz Sensing/Imaging

### 2.1. Classification of NDT Techniques in the Electromagnetic Spectrum

Figure 1 illustrates the position of microwave/terahertz-based NDT methods on the electromagnetic spectrum in relation to other common NDT techniques:

- Microwaves (µW): µW refers to electromagnetic waves with a frequency in the range of 0.3 to 300 GHz, corresponding to wavelengths of 1 m and 1 mm, respectively.
- Millimeter waves (mmW): The mmW frequency range spans from 30 to 300 GHz, corresponding to wavelengths of 10 mm and 1 mm, respectively. This technology is mainly applied in radar systems.
- Terahertz (THz) waves: THz waves are electromagnetic waves with a frequency ranging from 0.1 to 10 THz. They are located between the mid-infrared and microwave electromagnetic waves and have wavelengths from 3 mm to 30 µm.

The frequency $f$ and the wavelength $\lambda$ are related by the physical interrelationship $\lambda = c/f$, where $c = 299,792,458$ m/s is the speed of light in vacuum. Typically, higher frequency leads to increased 'resolution', enabling the detection of feature sizes using µW and THz techniques. Consequently, these methods can identify relatively small anomalies and flaws within the materials or objects being examined [10].

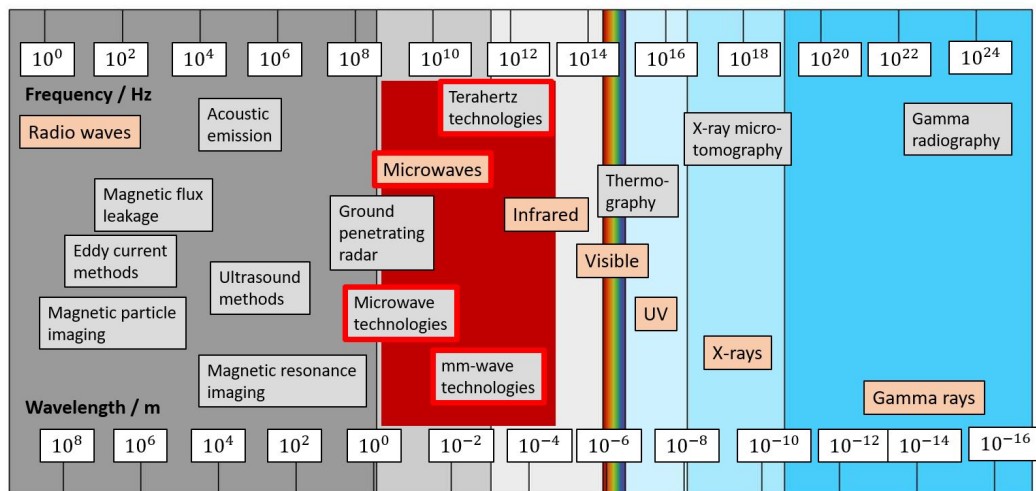

**Figure 1.** Microwave and terahertz-based techniques for NDT in the electromagnetic spectrum (highlighted in red; adopted from [10]).

### 2.2. Microwave Near-Field and Far-Field Imaging

Microwave imaging uses detection techniques to assess hidden or embedded objects in structures or media using electromagnetic waves in the microwave range from 0.3 to 300 GHz. Due to the geometric interdependence of the radiated electric and magnetic fields, MWI is broadly divided into near-field and far-field applications, each based on specific approaches and technologies. In general, the near-field is considered the region in the proximity of the sensor up to the Fraunhofer distance $r$, where

$$r = 2\frac{D^2}{\lambda} \tag{1}$$

$D$ is the spatial extent of the antenna and $\lambda$ the wavelength of the emitted signal. In this region, electromagnetic radiation has not developed yet and the resulting local electromagnetic field is in strong interference with adjacent loads and conductive elements. The electric and magnetic fields coexist independently of each other whereas in the far-field the electromagnetic radiation gradually develops and dominates. In contrast to the electromagnetic field, it is characterised by its transverse wave propagation.

In a near-field microwave imaging system, the aim is to detect the presence of an object or map electrical property distributions by measuring the scattered electric field from different positions [11], while taking into account the spherical wave propagation characteristics (see Figure 2). This involves placing multiple sensors typically in a coplanar alignment with respect to the main beam direction and close to the object while applying algorithms to the collected data for numerical analysis.

While the near-field imaging allows a high angular resolution for multistatic radar sensing by using the spatial diversity of the transmit and receive antennas, far-field imaging sets challenging requirements, especially for the hardware design, mainly due to the diffraction limit [12–14]. This also applies to the transmit power, too, which decreases with the square of the distance, and also requires lens with case-specific properties, which are required to influence the radiation pattern in order to establish focused propagating waves for sensitivity.

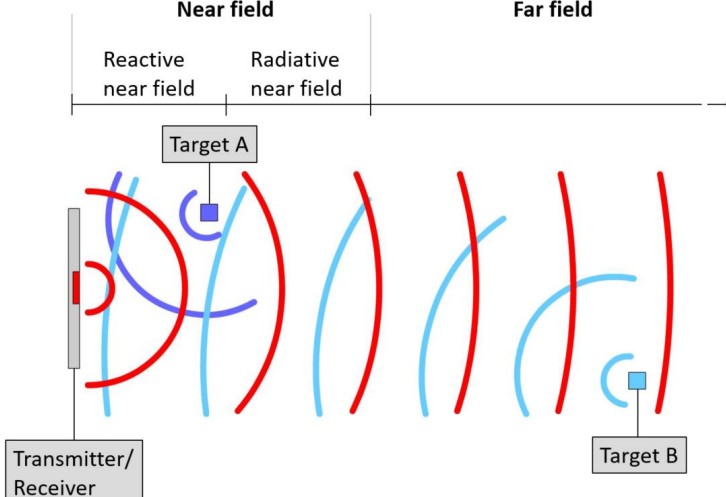

**Figure 2.** Wave propagation characteristics of emitted and reflected electromagnetic radiation between a sensor and a target in the reactive and radiative near field as well as in the far field region. The spherical waves gradually evolve into planar waves and the magnetic and electric fields of the electromagnetic radiation become stationary towards each other.

### 2.3. SISO and MIMO Radar Sensing

Conventional radar systems use single monostatic or bistatic antenna configurations for the transmission and receipt of RF signals, referred to as *Single-Input Single Output* (*SISO*). While the amplitude varies according to antenna characteristics, the emitted waveform remains constant, limiting signal processing methods to resolving distance, radial velocity (due to the Doppler effect) and, to some extent, angular information of targets. This is achieved by exploiting the amplitude and phase of the reflected signals. In addition to the limited beamforming capabilities, poor resistance to multipath propagation (clutter) results in a decreased low signal-to-noise ratio (SNR), which makes these systems unsuitable for near-field applications. One solution to this problem is to mechanically steer the sensor, whose orientation is known at all times, e.g., in airport surveillance radars (ASR).

*SIMO* (*Single-Input Multiple-Output*) radar systems can resolve target angles using arrays of receiving antennas. In principle, variations in signal phases together with their locations are used to determine the direction of the incoming signal, known as *direction of arrival* (*DoA*). In contrast, *MISO* (*Multiple-Input Single-Output*) systems use multiple transmit antennas and one receive antenna to enable multichannel signal processing.

*MIMO* (*Multiple-Input Multiple-Output*) radar systems are an upgradation to the aforementioned technologies. They consist of multiple equidistantly spaced transmit and receive antenna apertures typically in dense ULA (Uniform Linear Array)-like or sparse, colocated and coplanar arrangements, which create spatially varying time-dependent radiation patterns. This enables the use of virtual channels that exploit waveform orthogonality for encoding (time, frequency, or code; see Figure 3). Compared to SISO systems used in multipath scenarios, the channel capacity of this multistatic radar system is much higher and enables a robust and precise angle-specific target localisation using the spatial encoding, since the received signal (superposition of the reflected RF signals) is related coherently to the corresponding radiation pattern. Using powerful *digital beamforming* (*DBF*) algorithms, the three-dimensional reconstruction of the near-field scene from the back-scattered radiation is feasible and can be improved by algorithmic adaptations [15,16].

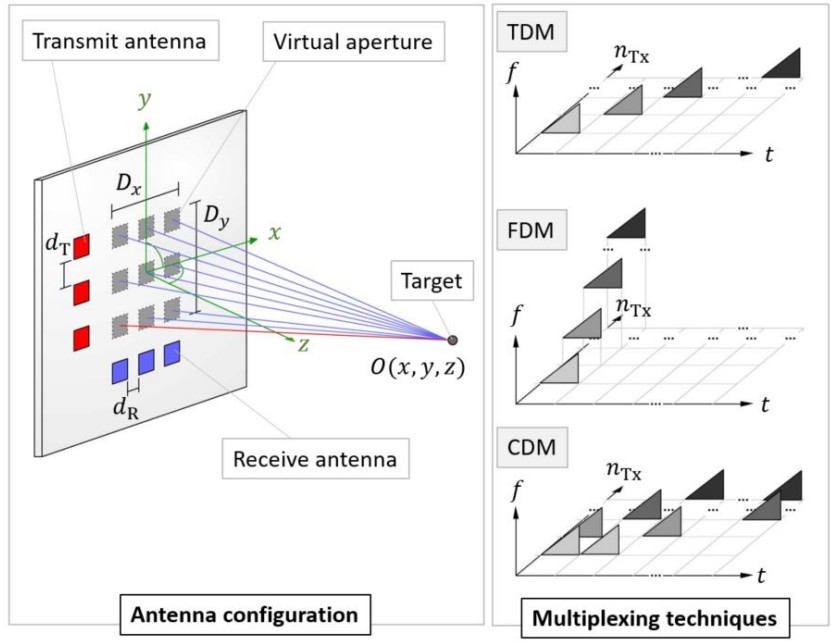

**Figure 3.** (**Left**): MIMO antenna configuration based on two orthogonal ULA for the transmit and receive antennas (red and blue, respectively) facing a point-like target; (**right**): Time-series of transmit antenna-specific signal frequency of linear FMCW-based MIMO sensors with respect to different multiplexing techniques.

### 2.4. Synthetic Aperture Radar Imaging

High-resolution imaging techniques require the use of larger apertures of transmit and receive antenna arrays, which is strongly linked to high costs for hardware. Synthetic aperture radar, which was invented in the early 1960s, provides a well-established viable solution for a variety of applications, such as satellite-based geological, maritime, agricultural or military imaging [17] without the necessity for large physical (real) apertures. Even with monostatic antenna configurations, with sufficient computing power and sophisticated signal processing techniques, it is possible to reach high spatial diversity for near-field and far-field applications. This is achieved by synthesising typically uniformly sampled RF baseband signal data as the radar or the target moves perpendicular to the direction of radiation using coherent processing. Efficient methods in the frequency domain, e.g., the Range Migration Algorithm (RMA) or the Back Projection Algorithm (BPA) [15], allow the precise determination of three-dimensional multiscattering environments using phase compensation to attenuate the influence of delays that were caused by the continuously changing distances between the radar and a specific point during measurement.

### 2.5. Terahertz Imaging

The development of terahertz-bases systems was significantly accelerated in the 1990s by the technological advances of signal generation based on short-pulsed lasers [18], while in recent years, the technical feasibility of economically efficient systems has been established after extensive research (*terahertz gap*). The fields of application are versatile and range from biomedical, agricultural, safety, to NDT in industrial processes [19].

The electromagnetic radiation is non-ionising (as it applies for the microwave domain, too) characterised by higher resolution in space and time. Despite the comparatively smaller penetration depth and susceptibility to attenuation by water, they are suitable for specific applications of NDT in food inspection.

Besides other classification types, i.e., based on the technological concept (e.g., THz time-domain spectrocopy, time-resolved THz spectroscopy, and THz emission spectroscopy as decribed in [19]), they can broadly be classified on the basis of their function principle (as in radar technology) namely into pulsed or CW-based systems.

Pulsed systems are based on transmission, reflection, or time-of-flight (ToF) modes. Time-domain spectroscopic systems (THz-TDS) are the most common type. These devices are operated to resolve data from the sample in the spatial or frequency domain based on transmission- or reflection-specific configurations. Systems operated in transmission mode are much simpler than in reflection mode since they do not require computationally-expensive calculation reconstruction and filtering techniques, which are necessary to mitigate the effects of polarization and refraction related to the incident angle.

For example, THz-TDS devices consist of a source, a beam splitter, an emitter, a detector, lenses for focusing, mirrors for steering, an adjustable delay unit, and a processing unit (computer). In order to create terahertz radiation, the source creates very short pulses (typically below one picosecond) using a titanium-sapphire laser that are split into a pump beam and the much weaker probe beam. The pump beam is directed to the emitter, which creates terahertz radiation using an antenna. The emitted radiation is collimated and directed using parabolic mirrors towards the sample. After passing through the sample it is convolved with the probe beam using a detector. During the measurement, the path of the pump beam is varied in order to create delays. This enables the spectrum of the electric field (amplitude) to be probed as a function of time, where the spectrum is obtained after applying Fourier transformation. This provides information about the dielectric properties, refractive index or the absorption coefficient of the material under test and enables valuable insight about the internal distribution of materials. Reflection-based systems vary with regard to the probe path, where the sample is radiated from a specific incident angle and the reflected radiation is collected from a symmetrically positioned collector. The time-of-flight (TOF) approach provides information about the internal structure using the round-trip time of the emitted pulsed RF signal [20].

In contrast, CW-based systems are comparatively much cheaper due to lower manufacturing costs of highly-integrated RF components and the commercial availability. However, systems that rely on measuring the scattering may require more time for the measurement due to computationally expensive signal processing operations, e.g., SAR-specific algorithms. This measurement requires either the sample or the sensor to be moved relative to each other, while the emitter sends and the collector receives continuous wave signals. The data acquisition process is incremental and local scatterers are determined after post-processing.

## 3. Review and Analysis of Related Work and Applications in Confectionary

Brinker et al. [10] reviewed recent advances in four major areas of microwave and millimeter-wave non-destructive testing and evaluation (NDT&E), namely materials characterization, surface crack detection, imaging and sensors. In food safety inspection, the detection of foreign bodies in various food products has been investigated. In previous reviews, Gowen et al. [18] and Mohd Khairi et al. [7] listed applications of non-invasive techniques for the detection of foreign objects in food. Only a few of them were based on µW/THz sensing techniques. In [21], Afsah-Hejri et al. reviewed various applications of THz spectroscopy and imaging in the food industry, such as foreign body detection, detection of toxic and harmful compounds, antibiotic detection, detection of microorganisms, moisture content measurement, inspection, identification, and quality control, adulteration detection, detection of carbohydrates and sweeteners, and detection of vitamins. A recent review on THz imaging for food inspection was presented by Zappia et al. [8]. In particular, they focused on transmission mode THz imaging, THz time-of-flight imaging and THz cameras. Nüßler and Jonuscheit [16] presented an overview of terahertz NDT methods as well as current and future industrial applications.

A more comprehensive and recent list of such applications for food safety and quality assessment, in particular for the detection of foreign objects in confectionery, found in publications during the last decade is summarised in Table 1.

Due to its high fat and low moisture contents, chocolate tends to be quite transparent to µW/THz energy. However, when foreign objects such as glass or plastic are introduced into chocolate, they change the scattering of a transmitted µW/THz wave, making their presence detectable [21]. This observation has been the basis of most work on the detection of foreign bodies in chocolate or similar food products. In [22,23], Jördens and Koch utilised a pulsed terahertz imaging system to detect foreign objects in chocolate. They used a single pulse structure to identify hazelnuts and a double pulse structure to detect non-metallic foreign objects, such as stones, glass, and plastic fragments, concealed within the chocolate bar. The presence of foreign bodies was evaluated based on their respective refractive indices, working within the integrated intensities between 0.4 and 0.75 THz. The study found that chocolate and hazelnuts share a refractive index of 1.75, while plastic fragments have a lower value of 1.5. In contrast, glass and stone have higher refractive indices of 2.6 and 1.9, respectively [7]. These varying refractive indices enable the effective differentiation and identification of these materials within the chocolate. Koch and Krok [24] used a similar THz-TDS system to examine pure and contaminated thick chocolate products. They stated that their system can also be used to inspect complex chocolate products such as waffles or truffles. Ung et al. [25] reported using THz-TDS imaging to detect contaminants in chocolate of varying thicknesses.

Phase differences in continuous wave (CW) systems allow for the measurement of various material parameters by leveraging the dielectric properties of materials. This capability to see through packaged products facilitates the detection of defects, such as missing pieces of chocolates, as demonstrated in the use case presented by Nüßler et al. [26]. They proposed measurement setups with near-field probes that provide good resolution combined with a compact design for low cost and easy integration into high-frequency systems up to 100 GHz. Their NDT technique was applied to the detection of a small piece of glass embedded in a (prepared) cookie. Both glass and chocolate exhibit similar

attenuation coefficients within the measured frequency range. However, the scattering effects at the boundary between glass and chocolate enable easy detection of contamination, but this effect is limited to homogeneous materials since local variations of density or local accumulation of ingredients of different dielectric properties change the propagation of the radiation in terms of amplitude and phase. This could lead to a deteriorated determination of the contaminants due to weaker contrasts [26]. For items like biscuits with mixed ingredients, distinguishing between desired and undesired components becomes exceedingly challenging when their attenuation coefficients and physical dimensions are alike. The distinct material constants of glass and chocolate allow the signal's propagation time to differentiate between them [26].

A sub-THz imaging system was developed in [27] using a transmission-type, CW-based method and a polygonal mirror scanning technique. The system utilizes a single aspheric f-theta lens and a polygonal mirror to produce a sub-THz line scan illumination. The applicability of this technology to food quality inspection was demonstrated by successfully testing some food samples, such as crickets in pasta flour and a chocolate bar with melted pieces.

Yu et al. [28] proposed a high-speed terahertz imaging setup, using a continuous wave at 0.3 THz, and investigated the presence of a caterpillar in a chocolate product placed on a conveyor belt moving at a speed of 72 m/min. They used an orthogonally polarised THz wave for real-time imaging and showed that the system effectively detected the caterpillar even though the chocolate was wrapped in paper.

In the study by Ok et al. [29], they developed a large scan area transmission imaging system operating at 140 GHz and tested the system to inspect various food products. This system used mechanical raster to scan through two optical head modules: a subwavelength beam focusing module and a transmitted beam detection module. The study showed that the system is effective in detecting defects, such as cracks and foreign objects, in flat food products like packaged chocolate bars and dried laver.

Foreign bodies can be detected simultaneously at the same location with sub-wavelength resolution without removing the wrapper. Ok et al. (2018) [30] presented a sub-THz imaging system (similar to the one proposed in [29]) with a mounted test object on motorised stages. The sub-wavelength focused beam scanning method was used to non-destructively examine a chocolate bar that was found to contain foreign bodies such as dried maggots, paper clips, and mealworms. The foreign bodies embedded in the chocolate bar were clearly identifiable, although the dried maggot inside the wrapper was not distinguishable in the image [30].

In their paper [31], Küter et al. described the SAMMI (Stand Alone Millimeter Wave Image) system, a rotating scanner operating in continuous wave mode at 90 GHz, and evaluated the system by detecting defects (Several plastic pieces of different sizes) in a chocolate bar. The chosen millimetre wave band is well suited for applications where automatic detection of deviations from a known good sample is critical. SAMMI can detect metallic chippings by analysing the amplitude response of the sensor and small dielectric impurities in materials by using phase information [31]. Results were presented as amplitude and phase deviations in the measurements, indicating that all plastic pieces were easily identified in the overlay.

SAMMI's ability to non-invasively check the completeness of packaged goods is demonstrated in [32]. The experiment used a commercially available chocolate advent calendar as the test object. Three individual pieces of chocolate were removed from the calendar before the measurement and the calendar was then closed again. The results obtained in the amplitude and phase images clearly showed the absence of the three missing pieces of chocolate inside the closed package. This demonstrated the system's ability to detect and visualise missing components even when the packaging was sealed.

Tobón Vásquez et al. [33] developed and assessed a prototype µW imaging device designed for detecting plastic or glass fragments within homogeneous food, specifically hazelnut-cocoa cream, stored in plastic or glass jars along a production line moving at a

speed of 3 jars/s (or 48 cm/s). Their µW imaging system concept involved two antennas positioned on either side of the production line, or alternatively, two synthetic antenna arrays flanking the food product in the middle. The detection concept utilises a µW tomography approach for processing data and classifying imaging results based on appropriate metrics.

The study by Ricci et al. [34] demonstrated the viability of a microwave-driven device fused with a neural pattern recognition network for instant food inspection. They used a detection method based on microwave sensing that exploits the dielectric contrast between the potential intruder and its surroundings. This microwave-based device was tested on an industrial food production line and demonstrated its ability to identify millimeter-sized intrusions made of materials such as plastic, glass or wood — classes typically undetectable by other inspection equipment.

In [35], the TeraSense Group offers commercial imaging systems for THz and sub-THz frequency ranges, i.e., 0.1–1.0 THz, for many potential applications. See [3] for some examples of non-invasive applications of TeraSense's THz food scanners, such as the detection of missing chocolate bars in a box and the detection of metal or plastic debris inside food packaging, i.e., a metal screw on a chocolate bar.

In [36], Shchepetilnikov et al. detailed the components and characteristics of the linear scanning system (280 GHz) of TeraSense, focusing on dynamic range, resolution and operating speed. They demonstrated the capabilities of this scanner by imaging real samples in scenarios typical of industrial nondestructive testing and security screening.

The potential of THz imaging to enhance food safety inspections by identifying both visible and hidden anomalies within food products has also been demonstrated by Zappia et al. [8,37]. They used the Zomega FiCO system and examined laboratory-prepared chocolate samples consisting of a plastic carrier filled with chocolate cream in different scenarios. Their results demonstrated the ability of THz imaging to detect a pistachio shell on the surface of the chocolate cream and to identify a metal hidden within the sample. In addition, filtered data presented in the form of THz radargrams showed the discrimination of two reflections from different materials. The studies [8,37] also highlighted the need for THz radargrams to detect and locate foreign objects such as plastic and metal fragments.

**Table 1.** Summary and analysis of published applications of NDT using microwave/terahertz sensing for the detection of foreign bodies in confectionery food items. (THz: Terahertz, CW: continuous wave, MW: microwave, SAR: synthetic aperture radar, MIMO: multiple input–multiple output).

| Ref. | Year | NDT | Frequency/Spectral Range | Inspection Tasks/Foreign Bodies | Food Products |
|---|---|---|---|---|---|
| [22,24] | 2006 | THz time-domain spectroscopy imaging | 0.4–0.5 THz | Detection of glass splinters, small stones, and metal screws | Chocolate bars |
| [23] | 2008 | Pulsed THz spectroscopic imaging | 0.4–0.75 THz | Detection of glass splinters, small stones, and metal screws | Milk/Haselnut chocolate bars |
| [26] | 2013 | CW Millimeter-wave imaging (SAMMI) | 78 GHz | Detection of glas and metal impurities | Chocolate cookies |
| [26] | 2013 | CW Millimeter-wave imaging (SAMMI) | 78 GHz | Completeness check (detection of missing chocolate pieces) | Packaged chocolate |
| [27] | 2015 | CW Sub-THz imaging | 0.21 THz | Completeness check (detection of melted pieces) | Chocolate bar |
| [28] | 2015 | CW THz imaging | 0.3 THz | Detection of caterpillar | Chocolate bar |
| [30] | 2018 | Raster-scanning CW THz imaging | 140 GHz | Detection of dried maggots, paper clips, and mealworms | Chocolate bar |
| [29] | 2019 | Large-scan-area Sub-THz imaging | 140 GHz | Detection of plastic, rubber, pepper seeds, and metal washers | Chocolate bars |
| [31] | 2018 | CW Millimeter-wave imaging (SAMMI) | 90 GHz | Detection of Detection of plastic flakes and pieces of different sizes | Chocolate bar |
| [33] | 2020 | MW imaging (two horn antennas) | 9.0–11.0 GHz | Detection of plastic or glass fragments | Plastic or glass jar with hazelnut–cocoa cream |
| [34] | 2021 | MW sensing (Antennas array) | 9.0–11.0 GHz | Detection of millimeter-sized intrusions (splinters of glass, wood or plastic, small pieces of jar caps) | Plastic or glass jar with hazelnut–cocoa cream |
| [32] | 2021 | CW Millimeter-wave imaging (SAMMI) | 78 GHz | Completeness check (detection of missing chocolate chips) | Chocolate advent calendar |
| [32] | 2021 | CW Millimeter-wave imaging (SAMMI) | 78 GHz | Detection of glas fragments in chocolate mass | Double cookies |
| [3] | 2022 | CW THz imaging | 0.1 THz | Completeness check (detection of missing candy bars) | Packaged chocolate bars |
| [3] | 2022 | CW THz imaging | 0.1 THz | Detection of metal or plastic debris, e.g., stick or screw | Packaged chocolate bar |
| [8,37] | 2021, 2023 | THz imaging (Zomega FiCo system [38]) | 0.08–3.0 THz | Detection of plastic and metal fragments | Chocolate cream in plastic support |

## 4. Overview of Non-Destructive Testing Prototypes and Industrial Products

### 4.1. Waveguide Systems

A series of comparatively low-cost, low-powered prototypes (called SAMMI), which are capable of real-time using millimetre waves (M-band) have been iteratively developed by Demming et al. [39], Küter et al. [31], and Schwäbig et al. [40]. Vertically positioned and collinearly aligned transmit and receive antennas that mounted on mechanically rotating discs operating at 78 GHz were used by Demming et al., while the sample is moved between them using a conveyor belt during the measurement (see Figure 4a). The determined amplitude and phase information obtained while moving on circular patterns is post-processed using a phase unwrapping method to avoid sudden jumps and, together with the amplitude distribution, is passed to a clustering method based on the Agglomerative Clustering and Ward's method for two-dimensional reconstruction of the permittivity of the sample.

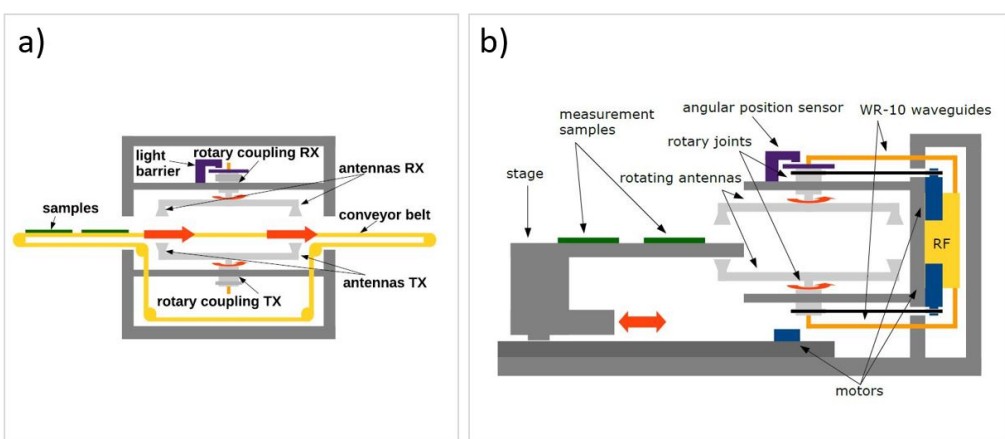

**Figure 4.** Mechanical concepts of SAMMI using transmitter and receiver in an opposing configuration, which are mounted on two synchronised rotating discs. The sample is located on the conveyor belt and probed in circular pattern: (**a**) first version operating at 78 GHz (Source: Schwäbig et al. [40] ©2021 Fraunhofer FHR); (**b**) second version using a 90 GHz CW system (Source: Küter et al. [31] ©2018 Fraunhofer FHR).

Küter et al. [31] improved SAMMI using a 90 GHz CW system and mitigated the disadvantages of the mechanical setup with regard to the maximum angular velocity of the rotating antenna setup, which was limited to 8 Hz due to the rotary joints (see Figure 4b). It has been increased to 15 Hz, giving a maximum of 80 Hz. The signal processing is implemented on a FPGA, which allows a high sample rate of 6.25 million samples/s. After filtering, the amplitude and phase are reconstructed by splitting the baseband signal into its in-phase and quadrature components.

Song et al. [41] focused on the development of an automatic detection approach using a THz-based imaging system based on transmission mode and operated at 0.1 THz that was attached to a conveyor belt. The high power source (up to 800 µW) provides the radiation, the intensity of which is measured by a 32 x 32 area scanner after being modified using a beam splitter and collimators and passing through the sample. Subsequently, the post-processed images were fed to deep learning-based pre-trained classifiers (ResNet50-Fast R-CNN and ResNet18-Fast R-CNN) for retraining and evaluation demonstrating their capability of yielding high classification performance (99.58 and 99.16 %, respectively).

### 4.2. SAR-Based Systems

Schwäbig et al. [40] extended the abilities of SAMMI (see Section 4.1) to resolve a three-dimensional (voxel-based) permittivity distribution using an efficient FMCW-based SAR. Due to novel approach, the mechanical setup was modified, where the transmit and receive antenna were placed both on the upper rotary disk (see Figure 5). The system

performs 192 sweeps during one semi-circular movement and the data are stored in a three-dimensional structure that is post-processed based on FFT interpolation and filtering to reduce spectral leakage. Using a modified version of the backprojection algorithm and two pre-computed matrices for the determination of distances and reachability between sensor and voxels, the data is converted to the frequency-domain and weighted with an exponential term, which contains the pre-computed values. After performing position corrections, the results are summed for each single sweep to obtain the final distribution of the permittivity. The signal processing is performed using a hardware concept, where the GPU is dedicated to process the image reconstruction operations taking advantage of its parallelization capability, while the CPU is used for data acquisition, data transfer, and visualization.

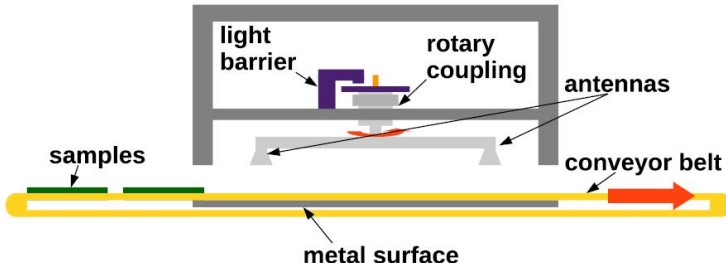

**Figure 5.** Mechanical concept of SAR-SAMMI. The sample is moved by the conveyor belt, while the antenna arm rotates. The transmit and receive antennas are placed on both ends of the rotor arm (Source: Schwäbig et al. [40] ©2021 Fraunhofer FHR).

A MIMO-SAR-based prototype operating in the upper sub-THz region (75–100 GHz) has been developed by [42]. The multistatic line arrays, consisting of sparsely distributed 12 transmit and 12 receive antennas, enable 3D-based image reconstruction for speeds up to 10 cm/s (see Figure 6). Three efficient signal processing techniques were implemented: Back-Projection (BP), Fast-Factorised Back-Projection (FFBP), and a modified algorithm based on the Range Migration Algorithm (RMA).

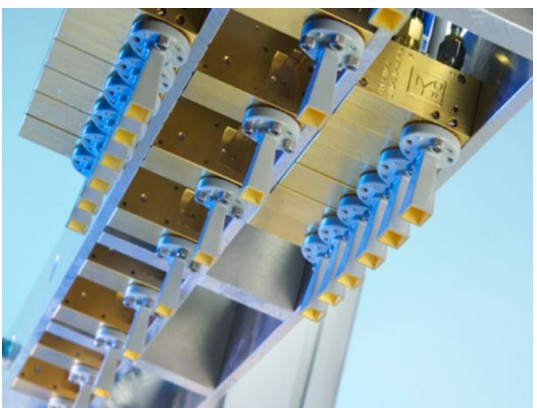

**Figure 6.** Linear antenna array configuration of a MIMO-SAR (Source: [42] ©2017 IEEE). The 12-element transmit antenna array lies between 2 receive antenna arrays, each equipped with 6 antennas (notice the varying distances along the transversal direction in order to obtain a specific virtual aperture based on spatial convolution).

### 4.3. VNA-Based Systems

A prototype for the inline detection of glass and plastic contaminants in hazelnut-cocoa cream glass jars has been developed by Tobón Vásquez et al. [33], based on a vector network analyser (VNA) using frequencies in the I/J-band (different configurations ranging from 9 to 11 GHz) that is connected to a ultra-wideband (UWB) transmit and a receive dual-ridged horn antenna, where the antennas are collinearly aligned towards

each other (see Figure 7). The VNA determines the $2 \times 2$ scattering matrix (S-matrix), which relates the complex outgoing RF signal to the incoming and, therefore, contains valuable information about the permittivity-related properties of the sample embedded in the amplitude and phase of the sampled signals. To create 2D images, the sample is moved at a constant velocity of 48 cm/s and, depending on the signal generation characteristics of the VNA and the geometrical properties of the selected sample, up to 13 positions can be sampled and used for image reconstruction. Contaminants are detected using a differential approach, where the deviation of the scattering parameters are determined with respect to a referential object using data of the determined electrical field and the truncated Singular Value Decomposition (TVSD) method to highlight deviation contrasts.

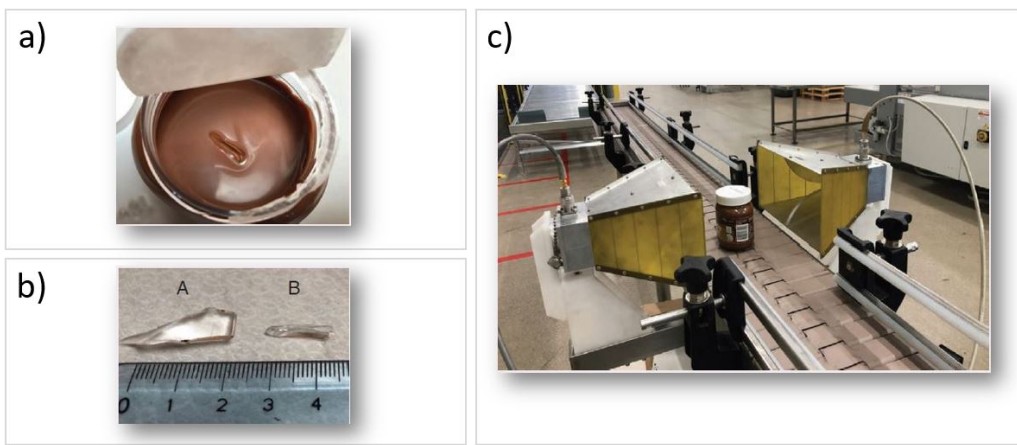

**Figure 7.** An experimental VNA-based system setup for measurements of jar-filled food products: (**a**) example of a contaminant within the hazelnut–cocoa cream jar; (**b**) two different plastic contaminants (A and B) considered; (**c**) configuration of opposing horn antennas and a jar moving on a conveyor belt (Source: [33] ©2020 IEEE).

Ricci et al. [34] have developed a microwave-based, real-time prototype using a VNA for a similar case, where the foreign object (splinters of glass, wood, or plastics) detection is applied to jars of hazelnut-cocoa cream, too. The multi-sensor array consists of six PCB antennas that were arranged in an arc shape, through which the sample passes during the measurement (see Figure 8). After data acquisition, which takes 50 ms to complete the scattering matrix, the data is collected and transferred to a neural network-based binary classification application. This concept was adopted later by Darwish et al. [43] to develop a Machine Learning-based classification method (based on SVM and MLP) to detect contaminants in an automated manner.

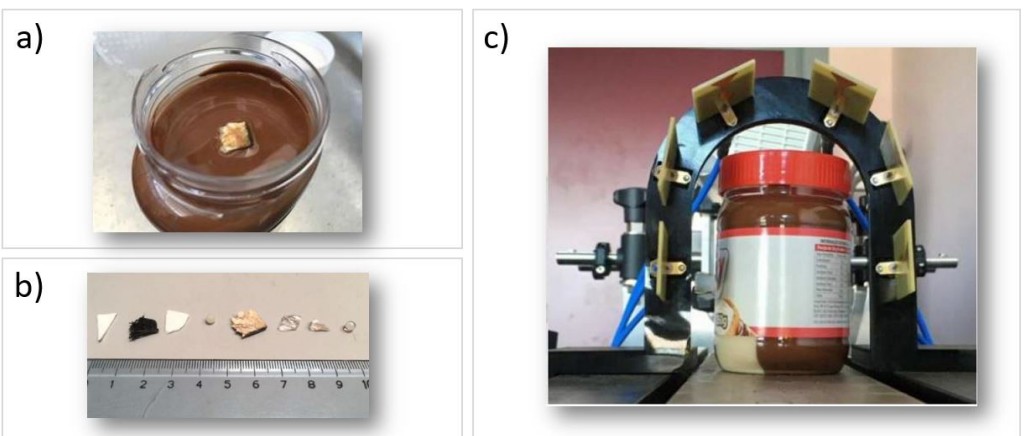

**Figure 8.** A VNA-based experimental setup for NDT of hazelnut-cocoa jars; (**a**) example of a contaminant within the hazelnut–cocoa cream jar; (**b**) the millimetre-sized foreign objects (glass, plastic, wood) considered; (**c**) An arc-shaped antenna array where the jar is moving in between with constant velocity (Source: [34] ©2021 IEEE).

### 4.4. Time-Domain Spectroscopy

A prototype terahertz-based spectroscopy system has been developed by Wang et al. [44], primarily for the detection of low-density contaminants. It consists of a time-domain spectroscope (Z3-XL THz-TDS) using a mode-locked femtosecond laser configured to operate at 80 MHz to generate 1560 or 780 MHz-based (switchable) pulsed optical signals (see Figure 9) that have a pulse width of less than 100 fs. Lenses are used to collimate the emitter and detector, which flank the sample during measurement. The sample is mounted on a motorised stage with two degrees of freedom, allowing a computer-controlled relative positioning with a lateral resolution of 0.1 mm.

The time-domain data are converted to frequency domain via FFT to obtain the electric field from which the amplitude is extracted and compared to a reference amplitude that was collected while the room was filled with nitrogen and without a sample. Finally, the resulting absorption coefficient and transmittance allow a two-dimensional image reconstruction after being processed by the Principal Component Analysis (PCA) to improve the data representation.

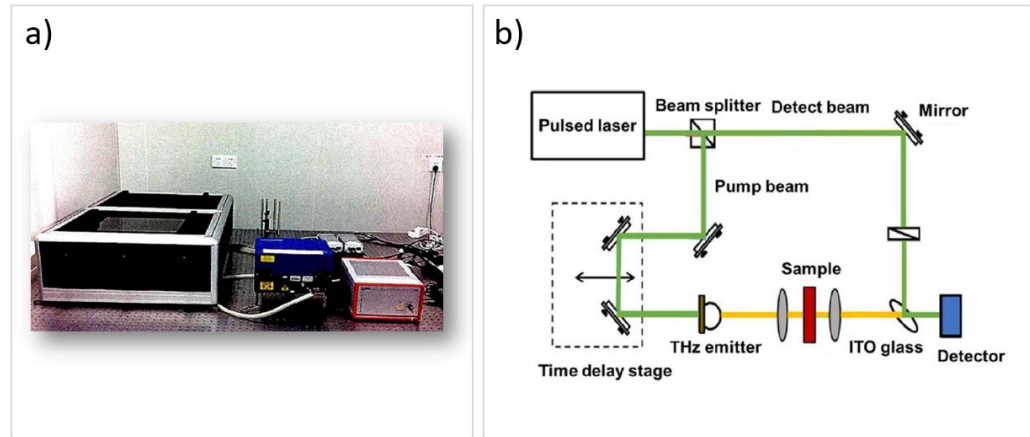

**Figure 9.** (**a**) Schematic of the THz spectroscopic imaging system; (**b**) System principle of the Z3-XL THz time-domain spectroscopy system (Zomega Corporation, East Greenbush, NY, USA), which operates on transmission mode (Source: [44] ©2020 Springer).

### 4.5. Gyrotron-Based Systems

Han et al. [45] have developed a compact gyrotron-based 2D imaging system at 0.2 and 0.4 THz for NDT (see Figure 10). A gyrotron uses a magnetron injection gun to create

a beam of electrons, which is compressed by a superconducting magnet, while being accelerated in a vacuum chamber, forcing the electrons to move along a helical path. The desired electromagnetic waves are formed by cyclotron resonance in the cavity resonator and propagate perpendicular to the electron beam, while the electrons are collected by a collector, minimising their remaining kinetic energy.

The comparatively high power electromagnetic radiation is directed towards the sample and collected by a pyroelectric array camera, which obtains the image based on the heat distribution of the detector. A static and an industrial prototype were developed using a square camera (12.4 mm² × 12.4 mm²) and a line scan camera (8 × 128 mm²), respectively, with the line scan camera providing an adequate frame rate of 24 Hz. The prototypes showed that, due to the comparatively wide beam width, a point-wise measurement is unnecessary since the high energy is distributed over a larger area, thus meeting the sensitivity requirements of the detector.

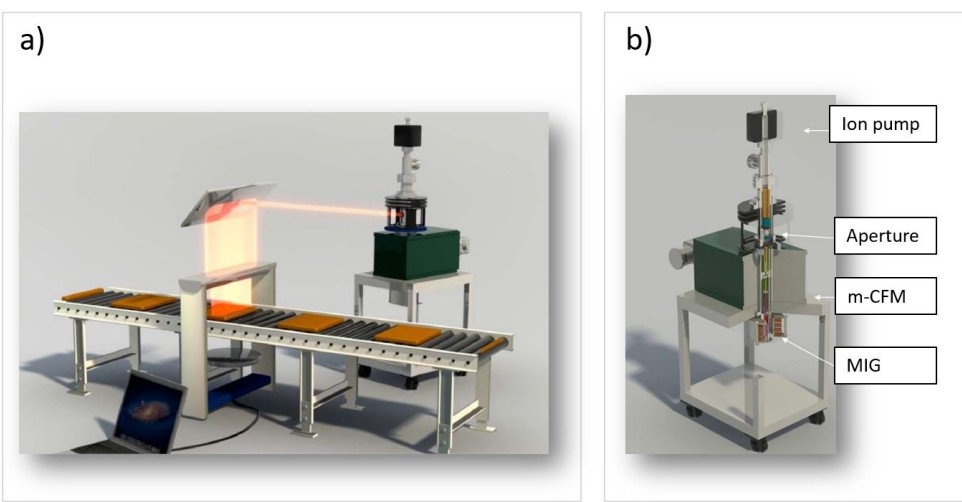

**Figure 10.** (**a**) Proposed configuration of sub-THz-based system named ARTIS for NDT in a production line. Gaussian beam from the gyrotron is transformed into a line shape crossing the conveyor belt after being deflected by a mirror to scan the product from above. A line array detector placed beneath take the shadow images of the samples moving along the conveyor belt; (**b**) System structure of the ARTIS (Source: [45] ©2020 IEEE).

*4.6. IMPATT Diode-Based Systems*

A real-time 100 GHz-based line scanner using high power IMPATT diodes has been developed by Shchepetilnikov et al. [46], which consists of a signal source unit that is attached to a horn antenna and the sensor array of the camera unit (detector; see Figure 11). The signal source unit is a compact device that provides a continuous wave signal of 100 GHz with an output power of 80 mW. Its power, frequency stability and ability to be synchronized with the camera provides high penetration capabilities. The camera's resolution is 1 × 256 px. and the high frame rate allows the processing of 5000 lines/second, enabling the industrial application in high-speed processing lines with speeds up to 15 m/s. During the measurement, the sample is moved through the source and detector as the systems acquires data, enabling 2D imaging based on the received power.

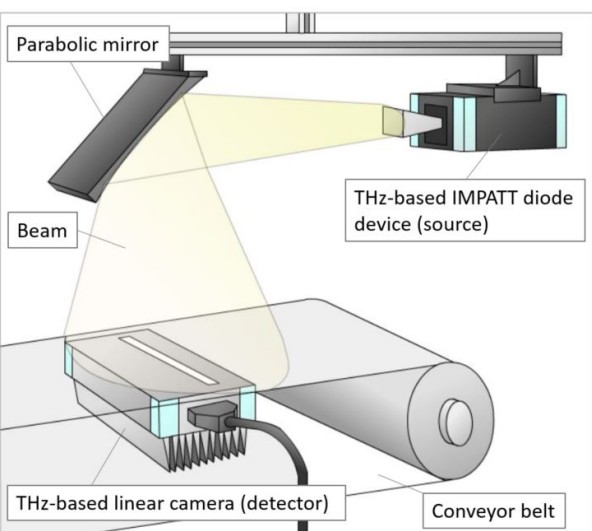

**Figure 11.** Principle of the experimental setup of Shchepetilnikov et al. [46] using the high-speed linear 100 GHz scanner. The radiation of the signal source (IMPATT diode device) is redirected towards the moving food sample (on a conveyor belt) from above using a concave mirror. The detector lies beneath the conveyor belt and is facing upwards.

## 5. Discussion

The work presented provides valuable insights into the versatility of microwave- and THz-based applications for inline food inspection and highlights the feasibility as well as the strengths and weaknesses of the specific technology and method using dedicated prototypes for studies under specific conditions. However, aspects of industrial suitability, practicality, reliability and accuracy need to be further explored to provide a common basis for evaluation and decision making for engineering purposes.

In industrial environments, the practical use of µW/THz NDT is assessed on parameters such as processing speed and required accuracy. Besides one-dimensional scanners [45], specific THz-TDS-based approaches require two-dimensional (grid-like) scan pattern combined with a specific beam width to increase the spatial accuracy, which is time-consuming and typically below processing line speeds (around 1.8 m/s). This aspect is discussed in some works [22,23] and solutions are provided using approaches, where the emitted waveform is sampled in a sub-region in an attempt to mitigate this disadvantage [22]. The work of Ok et al. [27] and Shchepetilnikov et al. [46] show that this problem also applies to microwave-based applications. In addition, the frequency-dependent beam width [23] has to be considered. Solutions, such as the work of Vásquez et al. [33], are able to cope with higher processing speed using fast SAR-based differential imaging methods.

The studies reviewed were based on non-standardised foreign bodies of different material, size, shape, location and orientation. For example, Ricci et al. [34] used millimetre-sized splinters of wood, glass and plastic (PTFE, nylon), while Jördens and Koch [23] used small stones, glass splinters, and metal screws. The influence of these particle properties on electromagnetic scattering and thus on the detectability, together with the inhomogeneity of the food sample, i.e., local variations in ingredients and consistency, especially water content, has not been thoroughly investigated. Preparation of the food sample, e.g., grinding, reduces the scattering, but requires destruction of the sample and makes NDT obsolete. In some of the works, validation was based on standardised objects (e.g., the 1951 USFA resolution target) [28,30]. This is appropriate for evaluating resolution, but does not reflect the requirements of industrial food production.

In addition, the production stage itself places limits on the practical use of µW/THz NDT systems. For example, the packaging of chocolate bars includes plastic, paper, or cardboard wrappings that are transparent to radiation in this frequency range, while aluminium foils pose a problem due to their impermeability. Applying the inspection in a

step prior to product staging and packaging (e.g., after the chilled chocolate bars have been demoulded and placed on the conveyor) would simplify the task, but would place spatial limitations on the equipment. Gyrotron-based inspection systems would be more affected than simpler microwave-based antenna arrays. Although this is much more relevant to engineering, it's not surprising that this aspect has not yet been addressed in related work, but remains a relevant issue.

Finally, the cost of THz-based inspection systems is also a critical issue limiting their dissemination, which has been highlighted in numerous works [8,18,21]. Microwave-based devices using low-cost PCBs [34] or SAR methods [40] are more compact and cheaper than gyrotron systems due to lower manufacturing costs for the low-cost highly-integrated ICs and materials, lower operating costs (gyrotrons require high power to build up the magnetic field or maintain active cooling) and lower technological complexity.

## 6. Emerging Research Topics and Future Application Directions

Although the effectiveness of μW/THz imaging has been demonstrated for many issues in food quality control, it is clear that there is still a long way to go before this technology can be applied in industrial processes.

- *Development of fast and economical μW/THz systems for inline food inspection*: The high cost of μW/THz instrumentation, coupled with penetration depth limitations and time-consuming processes, highlights the need for future research efforts. These efforts should aim to develop rapid and cost-effective THz systems by exploiting compact and more efficient equipment. This approach will facilitate wider accessibility and practicality in the use of μW/THz technology in various applications. The scanning speed needs to be improved in the future generations of THz systems. Systems in the €50k price range are necessary to cover the broad food quality control market.

  To reach this price range, transitioning from costly GaAs technology to the more affordable and commonly used silicon technology is essential. GaAs-based components are better suited for high-frequency technologies and applications compared to the silicon-based alternatives, but the material prices for this compound semiconductor are typically much higher, while silicon serves a general purpose in the electronic industry and is associated with low-energy consumption and lower manufacturing costs. This shift will empower us to merge THz detection and amplification circuits onto a single chip. This advancement will enhance pixel count and streamline the manufacturing process for chips and devices.

- *Introduction of a database library for food inspection*: A freely available database that contains data about the dielectric properties, e.g., transmission and absorption coefficients, refractive index, or the complex permittivity, of the basic food ingredients would contribute to the dissemination of material-specific knowledge and thus leverage the development of μW/THz systems as it is probably too complex to thoroughly examine all food products. Integration and commissioning would be accelerated by reducing time-consuming procedures for configuration and teaching through initial parameterisation. For this, the most important physical dependencies should be taken into account, e.g., thickness, density, temperature, hydration. In recent years, several related works have been carried out leading to the acquisition and collection of material-specific dielectric properties [21,47–49], but these studies have been carried out under constant conditions. In addition, carrying out measurements based on different pairings of the main food components and foreign objects that vary at least in material, shape, and size, providing data for creating domain-specific data representations, e.g., radargrams, would enhance the database and provide more insight into the interrelationships. Selection criteria for food products that are derived from the sample thickness, source power, frequency would introduce a unified approach towards the selection of inspection systems.

- *Need for more studies and industrial applications*: Although the potential of μW/THz imaging has been demonstrated for a number of use cases in food quality control,

as reviewed in Section 3 (Table 1), many more studies and industrial trials need to be carried out to precisely quantify how measurement conditions affect the accuracy and precision of µW/THz spectra and solutions. To date, no measurement trials or implementations in industrial plants have been found. Here, promising concepts where low-cost differential imaging systems, e.g., the system concept of [50], where symmetries of the product are exploited and do not require referential measurements should be pursued due to their practicability.

- *Development of machine/deep learning methods for automatic monitoring and recognition of food defects (foreign objects embedded in food)*: Design frameworks and pipelines for foreign body detection in food based on machine learning should be developed. Deep learning techniques, a powerful class of methods that can automatically learn feature representations from data, can be used in different architectures. The main challenge is to process the data and detect the foreign objects in real time and in high throughput food production lines. First ML-based MW sensing approaches for food contaminant detection have been proposed in [34,51], but these work directly on the raw MW signals and do not use the MW imaging data.

- *Development of hybrid solutions for machine vision and µW/THz systems for food inspection*: A combination of X-ray and vision inspection systems is currently being promoted as the ideal solution when products require vision inspection from above and/or below in addition to foreign body detection (see the system offered by WIPOTEC [52]). Future research and development should investigate a combination of µW/THz and vision inspection systems to take advantage of the µW/THz technology, in particular the improved characterisation of low density objects such as plastic, glass, wood or insects, and the use of harmless and safe non-ionising radiation since µW/THz technology does not pose any health risk to food or people like X-rays do.

## 7. Conclusions

This review provides an overview of the recent advances and developments in microwave- and THz-based Inline Food Inspection while setting emphasis on the packaged foods, i.e., confectionery products. The first section gives a brief description of the most common NDT methods that are classified within the electromagnetic spectrum. In the second section, the methods are narrowed down to the microwave and THz region, where techniques and domain-specific characteristics in terms of measurement principle and conditions are briefly explained, while the third section addresses recent works.

It can be concluded, that THz-based systems are superior in terms of scanning speed, resolution, and computational cost, which is confirmed by the studies using the prototypes reviewed in Section 4. While it has been observed that significant effort was put into developing powerful techniques for post-processing steps, such as denoising or multi-domain image reconstruction, there are physical limitations to overcome, with the penetration depth being the most critical, as voluminous or highly hydrated products could still pose a threat. Based on their comparably simpler technical complexity, microwave-based systems are substantially cheaper and easier to integrate. Their versatility and compact design make them suitable for various applications. SAR-based approaches using large subdivided virtual apertures and with application-specific adaptation of waveform patterns are the most promising in terms of resolution as they exploit the speed in processing lines.

Irrespective of the employed technology and methods, further studies need to be carried out, focusing on the creation of food-related databases, taking into account the different characteristics of food components as well as the characteristics of typical foreign objects and measurement conditions in order to facilitate the configuration and commissioning process. Nevertheless, microwave/THz-based NDT has proved its versatility and remains a promising technology for food quality assessment whose performance is expected to expand if sufficient emphasis is put on the main objectives.

**Author Contributions:** Conceptualization, M.J.; methodology, M.J. and K.P.; writing—original draft preparation, M.J. and K.P.; sketches and pictures, K.P.; writing—review and editing, M.J. and K.P.; manuscript revisions, M.J. and K.P.; supervision, M.J.; project administration, K.P. All authors have read and agreed to the published version of the manuscript.

**Funding:** This research received no external funding.

**Data Availability Statement:** Not applicable.

**Conflicts of Interest:** The authors declare no conflict of interest.

## Abbreviations

The following abbreviations are used in this manuscript:

| | |
|---|---|
| ASR | Airport Surveillance Radar |
| BPA | Back Projection Algorithm |
| CDM | Code Division Multiplexing |
| CW | Continuous Wave |
| DoA | Direction of Arrival |
| EM | Electromagnetic |
| FDM | Frequency-Division Multiplexing |
| FFT | Fourier Transform |
| FFBP | Fast-Factorised Back Projection |
| FMCW | Frequency-Modulated Continuous Wave |
| GHz | Gigahertz |
| IR | Infrared |
| MIMO | Multiple-Input Multiple-Output |
| MISO | Multiple-Input Single-Output |
| mmW | Millimeter Wave |
| MWI | Microwaving Imaging |
| NDT | Non-Destructive Testing |
| NIR | Near-Infrared |
| PCA | Principal Component Analysis |
| Radar | RAdio Detection and Ranging |
| RF | Radio Frequency |
| RMA | Range Migration Algorithm |
| SAMMI | Stand Alone Millimeter Wave Imaging |
| SAR | Synthetic Aperture Radar |
| SIMO | Single-Input Multiple-Output |
| SISO | Single Input Single-Output |
| TDM | Time-Division Multiplexing |
| TDS | Time-Domain Spectroscopy |
| THz | Terahertz |
| ToF | Time-of-Flight |
| SNR | Signal-to-Noise Ratio |
| VNA | Vector Network Analyzer |
| ULA | Uniform Linear Array |
| µW | Microwave |

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
