# Peer review of "Inline Inspection of Packaged Food Using Microwave/Terahertz Sensing—An Overview with Focus on Confectionery Products"

_processes, doi:10.3390/pr12040712_

Round 1

Reviewer 1 Report

Comments and Suggestions for Authors

Comments on the Quality of English Language

Minor editing is required.

Author Response

Dear reviewer,

Yours sincerely,
Mohieddine Jelali
Konstantinos Papadopoulos

Reviewer 2 Report

Comments and Suggestions for Authors

This paper deals with the exploitation of microwave/terahertz imaging technology for food quality control and assessment.Emphasis is placed on the (inline) inspection of wrapped/packaged food, such as chocolates, cookies, pastries, cakes, and similar confectionery products, moving along production conveyor belts.It describes in detail the state of development and the way forward.

I have the following comments on changes to this paper:

1.      In the figure1,you can make some changes to the figure to emphasize:Microwaves (µW)Millimetre waves (mmW)Terahertz (THz) waves,I can't tell which part you're emphasizing from Figure 1.This figure needs to be refined.

2.      Reasonably cite the content of the literature, and it is not advisable to pile up the literature in large volumes, and you must have your own opinions.

Comments on the Quality of English Language

1.      The 2 line in Abstract:change “an increased attention”to”increased attention”.

2.      The 283 line:change “scanning”to”to scan”.

3.      The 548 line:”for example”The grammar here is used incorrectly.

Author Response

(The authors gave the same response as above.)

Reviewer 3 Report

Comments and Suggestions for Authors

Very well written review article.

This is a good reference paper.

Comments on the Quality of English Language

- please review line 532 ("many" should replace "much")

- In the conclusion, "chapter" should be replaced by "section"

Author Response

(The authors gave the same response as above.)

Round 2

Reviewer 1 Report

Comments and Suggestions for Authors

Authors have revised the manuscript considering the suggestions/comments of the reviewers. In my opinion, it can be accepted for publication.